# Spatiotemporal patterns and entomological predictors of dengue transmission in Urban Surat, India (2016–2020): A surveillance-based risk modelling study

Jigna D. Gohil[1]*, Anjali M. Modi[2], Hiteshree C. Patel[1], Jayendra K. Kosambiya[3]

**1** Department of Community Medicine, Government Medical College, Surat, Gujarat, India, **2** Department of Community and Family Medicine, All India Institute of Medical Sciences (AIIMS), Rajkot, Gujarat, India, **3** Department of Community Medicine, Kiran Medical College, Surat, Gujarat, India

* johns.gohil12@gmail.com

## Abstract

Dengue fever is an increasing public health concern in urban India due to rapid urbanization, inadequate vector control, and underreporting of cases. Surat, a densely populated city in Gujarat, has remained a recurrent dengue hotspot, yet detailed spatiotemporal patterns and entomological determinants of transmission remain insufficiently explored. This study aimed to assess spatiotemporal patterns of dengue transmission in Surat and identify entomological and demographic predictors of elevated vector density to guide targeted urban interventions. A retrospective longitudinal analysis was conducted using five years of dengue surveillance data (January 2016–December 2020). A total of 1,658 laboratory-confirmed dengue cases reported to the sentinel surveillance system of the Surat Municipal Corporation were included. Monthly entomological surveillance assessed vector indices—House Index (HI), Container Index (CI), and Breteau Index (BI)—across administrative zones. Associations were examined using chi-square analysis, and binomial logistic regression identified predictors of moderate-to-high vector density (HI ≥ 1%) considering temporal, spatial, and demographic variables. Model performance was evaluated using receiver operating characteristic analysis. During the study period, reported dengue incidence declined from 11.1 to 2.2 cases per 100,000 population; however, elevated vector indices persisted, particularly in the South and South-East zones. Approximately 36% of cases occurred in households located in areas with HI ≥ 1%. Adolescents and young adults (median age 21.7 years) were disproportionately affected. Public sector facilities reported 2.6 times more cases than private providers (p < 0.001), suggesting substantial underreporting. Logistic regression identified year, month, zone, and age as significant predictors of elevated vector density (p < 0.001). The model showed moderate discrimination (AUC = 0.688), high specificity (87.3%), and low sensitivity (35.9%). Despite declining reported incidence, persistently high larval indices and

**Data availability statement:** The dengue surveillance dataset was obtained directly from SMC with formal permission for research use. As the data were not sourced from a public repository, the minimal anonymized dataset required to replicate the findings is provided as a Supporting Information file.

**Funding:** The author(s) received no specific funding for this work.

**Competing interests:** The authors have declared that no competing interests exist.

post-monsoon peaks indicate ongoing transmission risk, emphasizing the need for zone-specific vector control and strengthened surveillance systems.

## Author summary

Dengue, transmitted by *Aedes aegypti*, is a major public health concern in tropical cities, but understanding how disease risk relates to mosquito abundance and urban dynamics remains challenging. In Surat, India, we analyzed five years (2016–2020) of laboratory-confirmed reported Dengue cases alongside mosquito surveillance data to identify patterns of transmission, high-risk areas, and population groups most affected.

We found that reported Dengue cases declined sharply, particularly in 2020—likely due to COVID-19 disruptions—yet mosquito densities remained high in many parts of the city, showing that vector abundance alone does not predict disease incidence. Cases peaked during the post-monsoon months, and younger adults, especially males, were disproportionately affected. Spatial analysis revealed persistent hotspots in the South and South-East zones, while private healthcare facilities reported substantially fewer cases, highlighting surveillance gaps.

A predictive model for moderate-to-high vector density showed moderate accuracy, suggesting that integrating environmental, climatic, and socioeconomic factors could improve risk prediction. These findings provide actionable evidence for targeted vector control, improved disease surveillance, and public health strategies to reduce Dengue transmission in rapidly growing urban settings.

## Introduction

Dengue fever, caused by four distinct Dengue virus serotypes (DENV-1 to DENV-4), remains a major global public health threat. The World Health Organization (WHO) lists Dengue among the top ten global health threats, with an estimated 100–400 million infections annually and nearly half of the global population at risk [1–3]. Dengue disproportionately affects low- and middle-income countries, causing substantial morbidity, economic loss, and strain on health systems [1–3]. Asia accounts for over 70% of the global Dengue burden, with India contributing significantly to regional transmission [4].

In India, rapid urbanization, population growth, and environmental changes have intensified Dengue transmission in urban centers, where complex socio-ecological factors promote *Aedes aegypti* proliferation [4–7]. Unplanned urban expansion, inadequate water management, high human mobility, and climate variability amplify urban Dengue risk, linking prevention to global health priorities on climate resilience, sustainable urban development, and equitable health systems [1,2,6].

Surat, a rapidly expanding metropolitan city in Gujarat, is hyperendemic for Dengue, experiencing recurrent seasonal outbreaks and persistent viral circulation [8–11]. Its subtropical climate, with average temperatures of 25–35°C, mean annual rainfall of ~1,500 mm, and relative humidity consistently above 60%, is conducive to *Aedes aegypti* breeding and survival, the primary Dengue virus vector [11].

Dengue control is challenged by the absence of effective antivirals, limited vaccine efficacy and coverage, underreporting (particularly from private healthcare), and clinical overlap with other febrile illnesses, complicating early diagnosis and timely outbreak response [2,6,12]. Environmental determinants—including unplanned urban growth, suboptimal water management, high human mobility, and climatic variability—further exacerbate vector breeding and transmission [2,6,11].

*Aedes aegypti* is highly anthropophilic and breeds in diverse peri-domestic water containers. Its eggs remain viable during dry periods, resulting in spatially clustered transmission, particularly in densely populated urban areas with inadequate infrastructure [13,14]. This focal transmission underscores the need for fine-scale spatiotemporal surveillance to guide targeted interventions [13–15].

Entomological surveillance, using larval indices such as the House Index (HI), Container Index (CI), and Breteau Index (BI), is fundamental for assessing vector density. However, correlations between larval indices and laboratory-confirmed reported Dengue incidences are often weak or inconsistent, limiting predictive utility [6,15].

Surat's integrated vector-borne disease surveillance system, operational since 2006, consolidates laboratory-confirmed reported Dengue incidences with fortnightly entomological indices across administrative zones and healthcare providers [11]. This longitudinal dataset enables robust spatiotemporal analysis of Dengue transmission at fine geographic resolution. Yet, the predictive value of routine larval indices for identifying high-risk areas and periods remains inadequately quantified.

Research question: Can routinely collected entomological indices, combined with demographic, spatial, and temporal factors, predict moderate-to-high Dengue vector density in an urban hyperendemic setting?

Hypothesis: Moderate-to-high vector density (HI ≥ 1%) is significantly associated with specific demographic, spatial, and seasonal factors.

This study analyzes laboratory-confirmed reported Dengue incidences (2016–2020) alongside entomological surveillance data to identify spatial hotspots, seasonal trends, and demographic risk factors, and develops a logistic regression model predicting moderate-to-high vector density. Findings aim to inform targeted vector control interventions, improve early warning systems, and contribute a scalable predictive modeling approach for urban Dengue control in India and comparable low- and middle-income countries.

## Materials and methods

### Ethical considerations

Ethical approval was obtained from the Institutional Ethics Committee of Government Medical College, Surat (Approval No. 7319/21, dated 04/08/2021), with administrative clearance from the Surat Municipal Corporation. Given the secondary analysis of anonymized surveillance data, individual informed consent was waived.

### Study design and Setting

This retrospective surveillance-based study was conducted in Surat, a metropolitan city in Gujarat, western India, over a five-year period from January 2016 to December 2020. Surat had an estimated population of 4,645,384 based on the 2011 Census, projected using a decadal growth rate of 55.29% (approximately 4.5% annually). The city spans a geographical area of 462.15 km$^2$ with a population density of 10,052 persons per km$^2$ [16,17].

The study followed a post-positivist paradigm using a quantitative, retrospective longitudinal design. Programmatic cross-sectional dengue surveillance data collected over the five-year period were analyzed. All 1,658 laboratory-confirmed dengue cases reported during the study period were included.

The study was conducted in collaboration with the Vector-Borne Disease Control Department of the Surat Municipal Corporation (SMC), which functions as a central sentinel site under India's National Vector-Borne Disease Control Programme (NVBDCP) [8,11,16,17].

## Surveillance system and case notification (Fig 1)

Surat has a comprehensive healthcare infrastructure comprising 43 Urban Health Centres, 7 Community Health Centres, 2 government general hospitals, 2 municipal hospitals, 14 trust-run hospitals, 537 private hospitals, 1,515 private allopathic dispensaries, and 9 government dispensaries. Dengue is a legally notifiable disease in India [18]. Data collection is performed by Multipurpose Health Workers (MPHWs) who line-list suspected and laboratory-confirmed Dengue cases from both public and private healthcare facilities. These data are consolidated and validated at the SMC Central Sentinel Site [18].

Surveillance is implemented through two complementary mechanisms:

- Active surveillance: Conducted fortnightly by 489 trained surveillance workers performing house-to-house visits to identify febrile cases and inspect mosquito breeding sites.

- Passive surveillance: Routine reporting of suspected and confirmed Dengue cases—a notifiable disease—by Urban Health Centres and registered private providers.

## Case definition and laboratory confirmation

A confirmed Dengue case was defined as a clinically suspected case testing positive for NS1 antigen and/or Dengue-specific IgM antibodies, in accordance with NVBDCP case definitions. Laboratory testing employed MAC-ELISA kits supplied by the National Institute of Virology (NIV), Pune, and validated by the Indian Council of Medical Research (ICMR) [6]. Only laboratory-confirmed cases were included in the analysis.

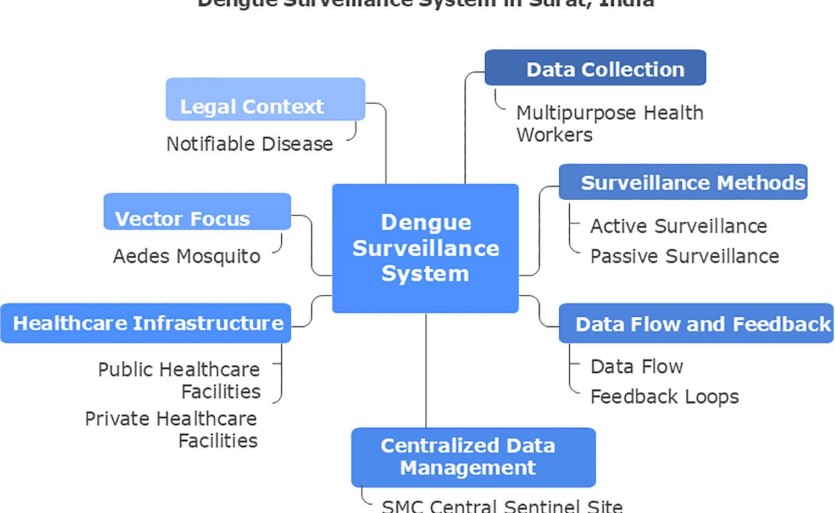

**Fig 1. Integrated urban Dengue surveillance framework in Surat, India.**

## Entomological surveillance and indices

Entomological surveillance was conducted fortnightly by trained anti-larval teams across all seven administrative zones of Surat. Vector infestation was assessed using standard larval indices for container breeders (*Aedes*), as per NVBDCP guidelines [2,5,12]:

• House Index: Proportion of houses inspected testing positive for mosquito larvae or pupae.

• Container Index: Proportion of water-holding containers positive for larvae or pupae.

• Breteau Index: Number of positive containers per 100 houses inspected.

These indices provide complementary measures of vector density: HI reflects household-level infestation risk, CI indicates container-specific infestation, and BI offers a density-adjusted measure of larval prevalence. NVBDCP thresholds consider HI ≥ 1%, CI ≥ 10%, or BI ≥ 5% indicative of elevated Dengue transmission risk [2,5,12].

HI ≥ 1% was used as the outcome in the logistic regression model, representing a programmatically actionable threshold for vector control and surveillance, per NVBDCP guidelines.

## Vector control interventions

The Surat Municipal Corporation implemented an Integrated Vector Management (IVM) strategy comprising continuous case detection and larval surveillance; larval control through source reduction, Temephos application, larvivorous fish deployment, and bio larvicides; adult mosquito control via targeted fogging and indoor residual spraying in high-risk areas; containment fogging within a 200-meter radius of confirmed cases; regulatory enforcement under the Bombay Provincial Municipal Corporation Act to prevent breeding violations; and community engagement through Information, Education, and Communication (IEC) campaigns conducted via schools, community outreach, and mass media.

## Data collection

The Surat Municipal Corporation has systematically collected Dengue surveillance data since 2007. For this retrospective study, data on 1,658 laboratory-confirmed Dengue cases reported between January 2016 and December 2020 were retrieved from SMC's Central Sentinel Surveillance database.

Extracted variables included date of case notification (month and year), residential address, patient age, sex, and entomological indices — House Index, Breteau Index, and Container Index. These variables were used to analyze spatiotemporal patterns and entomological predictors of Dengue transmission in urban Surat, India, as part of a surveillance-based risk modelling study. With administrative permission from the Health Commissioner, the dataset was accessed on 25 December 2021.

## Statistical analysis

All datasets were cleaned and screened for inconsistencies prior to analysis. Statistical analyses were performed using Jamovi software (version 2.6.144). Descriptive statistics summarized demographic and entomological variables, and chi-square tests assessed associations between categorical variables.

Spatial spot maps were generated using ArcGIS Online (free account) after extracting individual patients' latitude and longitude coordinates from their exact residential addresses, with base shapefiles for Surat Municipal Corporation obtained from DIVA-GIS (https://www.diva-gis.org/gdata, CC-BY 4.0).

## Case notification rate calculation

Annual case notification rates per 100,000 population were calculated using mid-year population projections based on the 2011 Census and an estimated annual growth rate of 4.5% [16]. Sector-specific rates (public vs. private) and zone-wise estimates were derived using fixed 2011 zonal population figures due to the unavailability of updated projections [17].

## Binary logistic regression model

A binomial logistic regression model was used to identify predictors of moderate-to-high vector density (HI ≥ 1%) in accordance with NVBDCP risk thresholds [5,6]. Areas were classified as low-risk (HI < 1%) or moderate-to-high risk (HI ≥ 1%).

Predictor variables included age (continuous), sex, year, month, and residential zone (categorical). Multicollinearity was assessed using Pearson's correlation coefficients, variance inflation factors (VIF), and tolerance values, with no violations observed (VIF < 1.2; tolerance >0.8). Model fit was evaluated using deviance, Akaike Information Criterion (AIC), and pseudo-$R^2$ statistics (Cox & Snell, McFadden).

Model performance was evaluated using receiver operating characteristic (ROC) curve analysis, reporting accuracy, sensitivity, specificity, and area under the curve (AUC) to support operational relevance for vector control planning.

## Results

### Overall dengue burden and laboratory confirmation

Between January 2016 and December 2020, a total of 1,658 laboratory-confirmed dengue cases were reported in Surat through the central sentinel surveillance system, encompassing notifications from both public and private health facilities. Diagnostic confirmation was achieved through detection of the dengue non-structural protein 1 (NS1) antigen in 66.0% of cases, dengue-specific immunoglobulin M (IgM) antibodies in 44.7%, and both markers in 10.8% of cases.

### Temporal distribution of dengue cases

The annual notification rate of reported Dengue incidences in Surat demonstrated a declining trend over the five-year period, with an overall rate of 5.44 cases per 100,000 population. The highest annual notification rate was observed in 2016 (11.1 per 100,000), followed by a steady decline to 2.2 per 100,000 in 2020 (Fig 2; Table A in S1 Text).

The annual distribution of reported Dengue incidences was 2016 (37.3%), 2017 (16.4%), 2018 (15.0%), 2019 (22.6%), and 2020 (8.6%) (Fig 2, Table A in S1 Text). Aggregated by month, incidences showed a distinct seasonal pattern, with September (22.1%), October (24.1%), and November (20.3%) accounting for the majority, while February (0.5%) and April (0.4%) had the fewest reported Dengue incidences. The post-monsoon period (September–October) accounted for 46% of total reported Dengue incidences.

### Entomological indices and transmission

Entomological surveillance revealed fluctuations in vector indices over the study period. The House Index was ≤ 1 in 1,053 (64%) of surveyed areas; however, 12 houses (0.729%) had a House Index between 5 and 10, and 2 houses (0.1215%) exceeded a House Index of 10. The Container Index remained ≤10% across all surveyed locations. The Breteau Index was ≤ 5 in 1,632 cases (99.1%) (Fig 2, Table A in S1 Text).

Significant interannual variability in House Index and Breteau Index was detected (p < 0.001) (Table A in S1 Text). Notably, the year with the highest vector indices (2017) did not correspond to increased reported Dengue incidence.

### Spatial distribution of reported dengue incidence and entomological risk

Marked spatial heterogeneity in Dengue notification was observed across Surat's administrative zones. The South-East zone reported the highest Dengue notification rate (55.1 per 100,000 population), followed by the South zone (40.1 per 100,000) (Fig 3).

The contribution of the South zone to total reported cases increased substantially over time, rising from 17% in 2016 to 29% in 2020. In contrast, the Central and West zones experienced notable declines, with case proportions decreasing from 10.2% to 2% and from 17% to 0.7%, respectively (Fig 4; Table B in S1 Text).

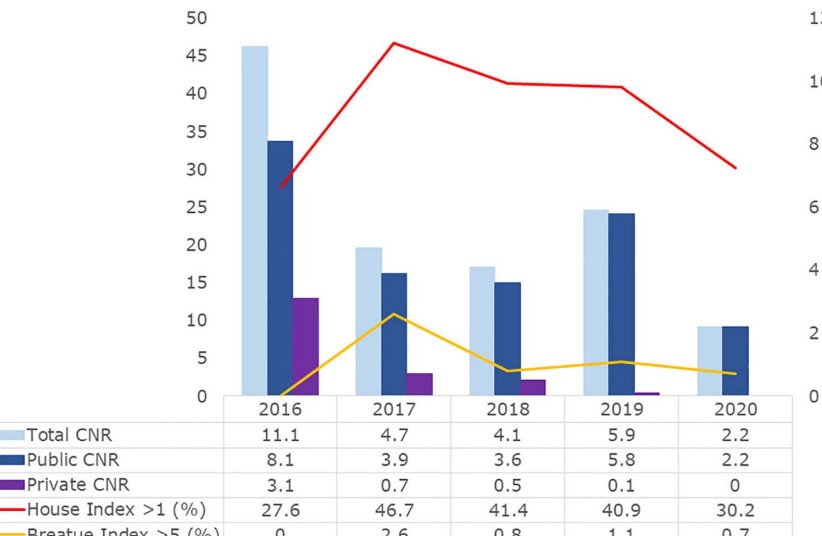

| | 2016 | 2017 | 2018 | 2019 | 2020 |
|---|---|---|---|---|---|
| Total CNR | 11.1 | 4.7 | 4.1 | 5.9 | 2.2 |
| Public CNR | 8.1 | 3.9 | 3.6 | 5.8 | 2.2 |
| Private CNR | 3.1 | 0.7 | 0.5 | 0.1 | 0 |
| House Index >1 (%) | 27.6 | 46.7 | 41.4 | 40.9 | 30.2 |
| Breatue Index >5 (%) | 0 | 2.6 | 0.8 | 1.1 | 0.7 |

**Fig 2. Annual temporal distribution of laboratory-confirmed reported Dengue incidences and entomological indices in urban Surat, 2016–2020.**

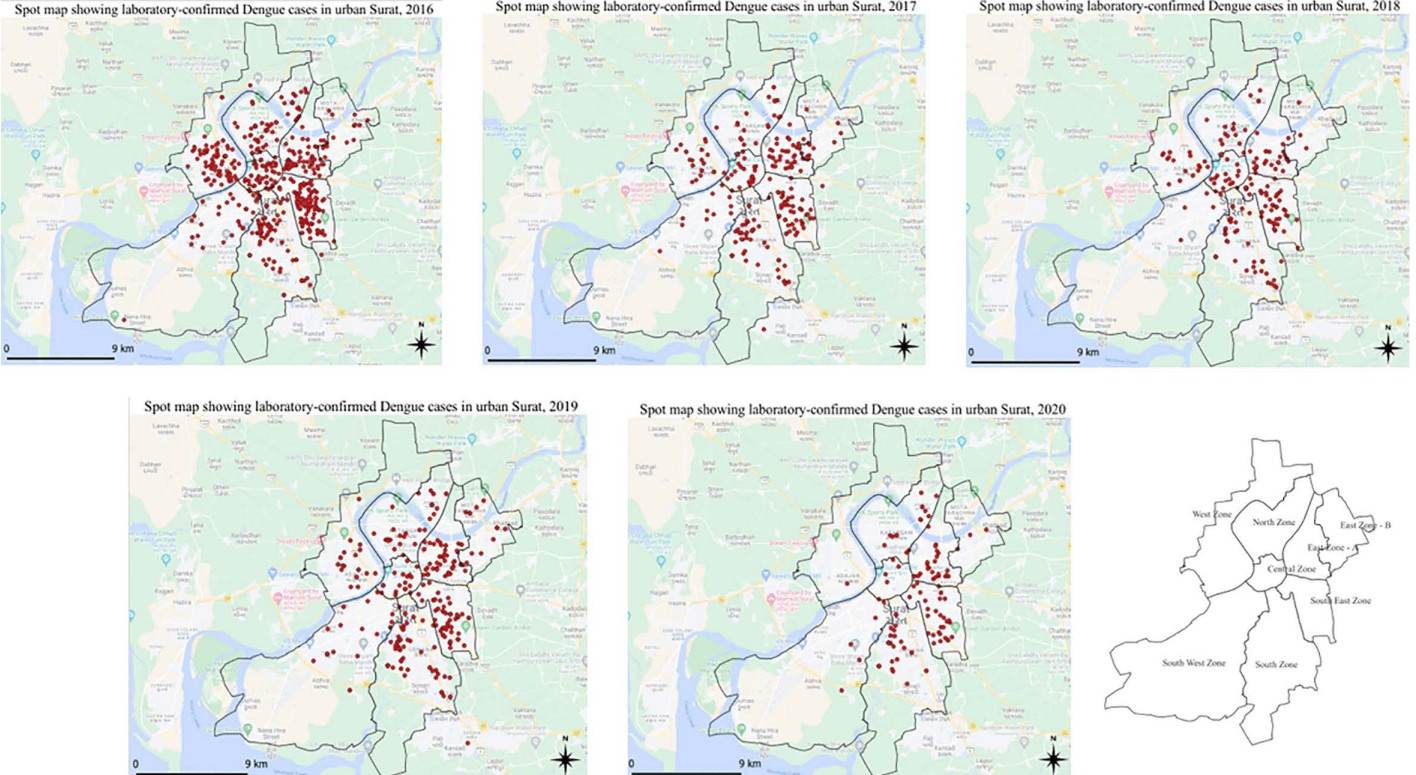

**Fig 3. Spatial distribution of laboratory-confirmed reported Dengue incidences in urban Surat, 2016–2020.** (Case coordinates were mapped in ArcGIS Online using the Surat Municipal Corporation boundary shapefile from DIVA-GIS).

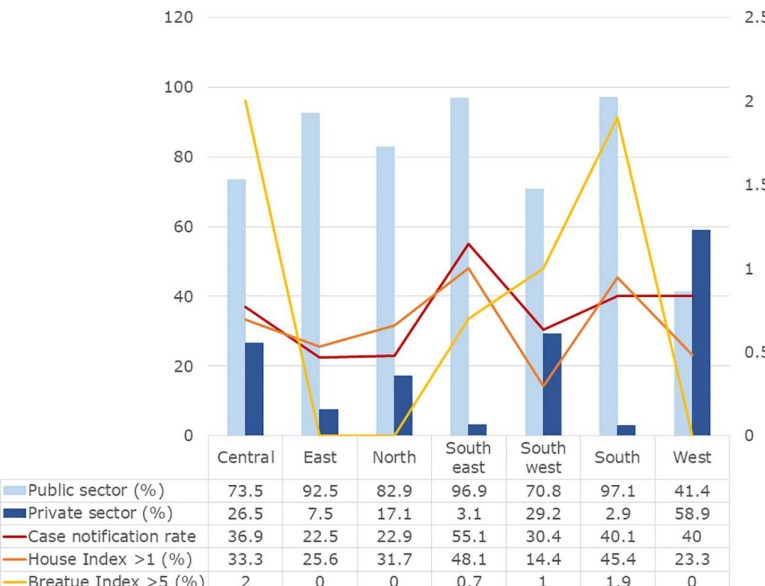

| | Central | East | North | South east | South west | South | West |
|---|---|---|---|---|---|---|---|
| Public sector (%) | 73.5 | 92.5 | 82.9 | 96.9 | 70.8 | 97.1 | 41.4 |
| Private sector (%) | 26.5 | 7.5 | 17.1 | 3.1 | 29.2 | 2.9 | 58.9 |
| Case notification rate | 36.9 | 22.5 | 22.9 | 55.1 | 30.4 | 40.1 | 40 |
| House Index >1 (%) | 33.3 | 25.6 | 31.7 | 48.1 | 14.4 | 45.4 | 23.3 |
| Breatue Index >5 (%) | 2 | 0 | 0 | 0.7 | 1 | 1.9 | 0 |

**Fig 4. Zone-wise entomological indicators of laboratory-confirmed reported Dengue incidences in urban Surat, 2016–2020.**

Entomological risk mirrored these spatial patterns. An HI > 1 was recorded in 48.1% of surveyed areas in the South-East zone and 45.4% in the South zone. Among the 14 areas (0.9%) where BI exceeded 5, half were located in the South zone. Spatial variation was statistically significant for House Index ($\chi^2 = 88.1$, df = 6, N = 1,646; p < 0.001), while variation in Breatue Index did not reach statistical significance (p = 0.053) (Table B in S1 Text).

### Demographic characteristics of reported dengue incidences

The median age of confirmed reported Dengue incidences cases was 21.7 years (interquartile range: 13–29), indicating a higher burden among adolescents and young adults. Age was inversely correlated with all entomological indices: House Index: r = -0.138, p < 0.001; CI: r = -0.137, p < 0.001; BI: r = -0.115, p < 0.001. The male-to-female ratio was approximately 3:2, although no significant differences in entomological indices were observed between males and females.

### Healthcare sector reporting patterns

Of the 1,658 reported Dengue cases, 85.0% (n = 1,409) were reported through public healthcare facilities. The Dengue notification rate from public institutions was 2.6 times higher than that from private providers. Private-sector reporting declined sharply over the study period, falling from 8.1 to 0 cases per 100,000 population between 2016 and 2020.

In the South and South-East zones, more than 96% of cases were reported by public facilities. In contrast, relatively higher proportions of private-sector reporting were observed in the West (59%), South-West (29%), and Central (27%) zones, although these contributions declined markedly after 2018.

### Predictors of entomological risk

A binary logistic regression model was fitted to identify demographic, temporal, and spatial predictors of moderate-to-high vector density, defined as House Index ≥1%. The model demonstrated good overall fit (Deviance = 1975; AIC = 2025), with

pseudo-$R^2$ values indicating moderate explanatory power (Cox & Snell $R^2$ = 0.0823; McFadden $R^2$ = 0.140). No multicol-linearity was observed among predictors, with all pairwise correlations <0.7 and VIFs ranging from 1.01 to 1.19.

Age was a significant negative predictor of vector density (OR = 0.98; 95% CI: 0.97–0.99; p<0.001). Compared with 2016, significantly higher odds of House Index ≥1% were observed in 2017 (OR = 2.44), 2018 (OR = 1.73), and 2019 (OR = 2.05) (all p<0.001), while 2020 showed no significant difference (Table 1).

Several months showed significantly higher odds of moderate-to-high vector density relative to January, particularly August (OR = 6.025, p=0.002), July (OR = 5.434, p=0.006), September (OR = 5.638, p=0.003), and December (OR = 3.335, p=0.045).

Spatially, residents of the East (OR = 0.416, p<0.001), South West (OR = 0.216, p<0.001), and West zones (OR = 0.407, p<0.001) had significantly lower odds of being in high-risk areas compared to those in the South zone (Table 1).

## Model performance

The Binary logistic regression model demonstrated an overall classification accuracy of 68.8%, with high specificity (87.3%) and moderate sensitivity (35.9%). The logistic regression model demonstrated an overall classification accuracy of 68.8%, with high specificity (87.3%) and moderate sensitivity (35.9%). The area under the receiver operating characteristic (ROC) curve was 0.687 (Fig 5).

**Table 1. Binary logistic regression predicting moderate-to-high vector density (HI ≥ 1%) among laboratory-confirmed reported Dengue incidences, Surat, 2016–2020 (N = 1646).**

| Predictor | Estimate | p-value | Odds Ratio (95% Confidence Interval) |
|---|---|---|---|
| Intercept | −1.7220 | 0.005 | 0.18 (0.05 − 0.59) |
| Age (years) | −0.0209 | <0.001 | 0.98 (0.97 − 0.99) |
| Gender (Female vs Male) | 0.0582 | 0.603 | 1.06 (0.85 − 1.32) |
| Place (Government vs Private) | 0.0338 | 0.861 | 1.03 (0.71 − 1.51) |
| Year (reference: 2016) | | | |
| 2017 | 0.8936 | <0.001 | 2.44 (1.78 − 3.35) |
| 2018 | 0.5492 | <0.001 | 1.73 (1.25 − 2.40) |
| 2019 | 0.7182 | <0.001 | 2.05 (1.51 − 2.78) |
| 2020 | 0.1520 | 0.488 | 1.16 (0.76 − 1.79) |
| Month (reference: January) | | | |
| August | 1.7960 | 0.002 | 6.03 (1.92 − 18.92) |
| July | 1.6926 | 0.006 | 5.43 (1.64 − 17.99) |
| September | 1.7295 | 0.003 | 5.64 (1.82 − 17.47) |
| December | 1.2045 | 0.045 | 3.34 (1.03 − 10.83) |
| March | 1.5883 | 0.049 | 4.90 (1.01 − 23.78) |
| June | 1.4377 | 0.042 | 4.21 (1.05 − 16.83) |
| October | 1.3162 | 0.023 | 3.73 (1.20 − 11.56) |
| November | 1.2776 | 0.027 | 3.59 (1.16 − 11.14) |
| Zone (reference: South) | | | |
| East | −0.8767 | <0.001 | 0.42 (0.29 − 0.60) |
| South West | −1.5307 | <0.001 | 0.22 (0.12 − 0.40) |
| West | −0.8993 | <0.001 | 0.41 (0.26 − 0.64) |
| Central | −0.5089 | 0.018 | 0.60 (0.39 − 0.92) |
| North | −0.5367 | 0.009 | 0.59 (0.39 − 0.88) |
| South East | 0.1349 | 0.368 | 1.14 (0.85 − 1.54) |

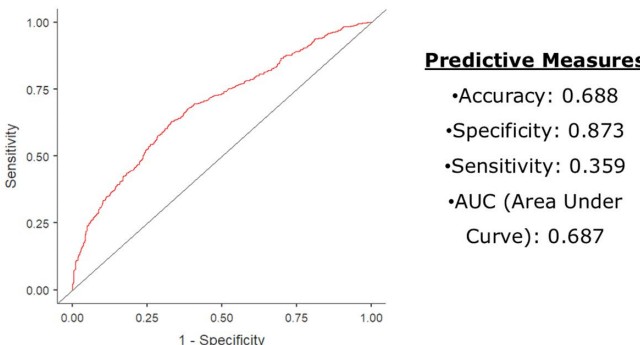

**Predictive Measures**
- Accuracy: 0.688
- Specificity: 0.873
- Sensitivity: 0.359
- AUC (Area Under Curve): 0.687

**Fig 5. ROC analysis and predictive accuracy of the binary logistic regression model for moderate-to-high vector risk (House Index ≥1%).**

## Discussion

This study provides critical insights into reported Dengue incidence and vector risk in Surat, India, between 2016 and 2020 by integrating laboratory-confirmed reported Dengue cases with entomological indices, demographic variables, and spatial data. The results refine our understanding of Dengue transmission dynamics and *Aedes aegypti* ecology in a rapidly urbanizing Indian city, highlighting the complex interactions between vector ecology, human exposure, and urban surveillance systems.

### Temporal trends and the role of public health infrastructure

Reported Dengue incidence declined from 11.1 to 2.2 cases per 100,000 population over the study period. This decline contrasts with national and regional trends of stable or increasing Dengue burden and should therefore be interpreted cautiously (Fig 6) [7,9–11,19,20]. Dengue surveillance is highly sensitive to reporting practices, diagnostic availability, and health system disruptions, particularly in urban settings [1,15,21,22]. The pronounced drop in 2019–2020 likely reflects underreporting associated with the COVID-19 pandemic, including reduced health service access and diversion of resources [23]. Consequently, observed temporal patterns primarily reflect fluctuations in reported cases rather than actual transmission intensity [1,23,21].

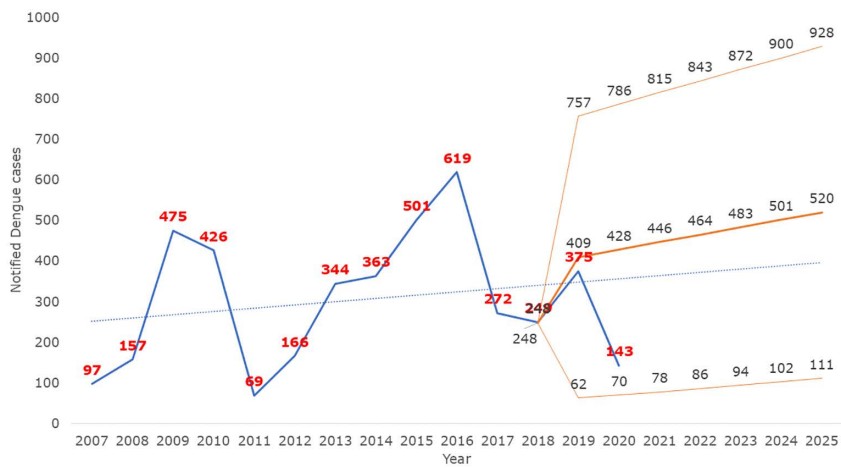

**Fig 6. Time trend of laboratory-confirmed reported Dengue incidences in urban Surat, 2007–2025.**

## Private-sector reporting gaps and policy implications

Dengue surveillance in Surat combines active surveillance—fortnightly house-to-house visits by public-sector workers—and passive surveillance, the routine reporting of suspected and confirmed cases by Urban Health Centres and private providers. Despite the predominance of private facilities, underreporting from this sector remains a critical gap, driven by variable compliance, limited laboratory confirmation, administrative burden, lack of incentives, and weak enforcement of notification mandates [9,24,25]. These gaps worsened during the COVID-19 pandemic [23]. As active surveillance relies on public infrastructure, the dominance of public-sector notifications reflects system design rather than true healthcare utilization [18,24]. Strengthening urban Dengue monitoring requires mandatory reporting across all sectors, interoperable digital systems, and ongoing engagement and training of private practitioners [2,6,24,21,26].

## Seasonality and the timing of interventions

Nearly half of the reported Dengue cases occurred during the post-monsoon months of September and October, coinciding with increases in vector density following monsoon rains [6,9,10]. This consistent seasonal pattern supports the strategic intensification of vector control measures ahead of and during high-risk months, including larval source reduction, targeted fogging, and community education, coupled with improved urban water and waste management [2,5,13].

## Spatial heterogeneity and targeted control

Spatial clustering of cases in the South and South-East zones, which also exhibited the highest vector indices, is consistent with known associations between urban density, inadequate infrastructure, and Dengue risk [9–11,27,28]. Mapping these hotspots supports geographically focused interventions—optimizing the allocation of surveillance, source reduction, and health communication efforts [27].

## Demographic risk profile

The median age of Dengue patients was 21.7 years, and the male-to-female ratio was approximately 3:2—patterns frequently observed in other urban Dengue settings attributable to greater exposure among younger, more mobile males [7,14]. Public health education targeting adolescents and young adults, via educational institutions and community outreach, should emphasize use of protective measures (e.g., repellents, protective clothing) and active participation in household-level vector control [7,14].

## Entomological indices and transmission decoupling

Despite reductions in reported Dengue incidence, entomological surveillance showed sustained vector presence, with high larval indices poorly corresponding to case notifications, particularly in 2017 [10,15,29]. Larval indices (HI, CI, BI) are limited predictors of Dengue risk, as they do not capture adult female mosquito density, vector competence, human–vector contact, or viral circulation [15,21,29,30]. In the absence of adult vector, climate, and mobility data, larval indices should be interpreted as indicators of vector presence rather than transmission intensity [6,15,21,30]. Although correlations with reported incidence are weak, integrated epidemiological and entomological surveillance remains essential, as single data streams cannot provide reliable early warning [6,15,21]. Integration improves situational awareness and preparedness, rather than precise outbreak prediction, aligning with WHO guidance and urban Dengue frameworks [6,21,26].

## Predictive modelling of vector risk

The logistic regression model demonstrated moderate discrimination (AUC = 0.687), with high specificity (87.3%) but low sensitivity (35.9%), explaining 14% of the variance. The low sensitivity likely reflects the absence of key determinants of

Dengue risk, including rainfall variability, temperature, housing conditions, water storage practices, population mobility, and socioeconomic factors such as migration and urban poverty [6,15,27,21,31].

Accordingly, the model should be interpreted as a screening or prioritization tool, effective for identifying lower-risk areas but not for precisely predicting Dengue hotspots [15,21].

### Limitations and strengths

Several limitations should be acknowledged. First, underreporting—particularly from private healthcare providers and during the COVID-19 period, as well as undiagnosed infections—of dengue cases reported to the sentinel surveillance site likely resulted in underestimation of the true dengue incidence. Second, reliance on larval indices limits inference regarding adult mosquito abundance and actual transmission potential. Third, the predictive model did not incorporate climatic, environmental, or socioeconomic variables, thereby constraining its explanatory power.

Despite these limitations, the study has notable strengths, including the use of laboratory-confirmed dengue cases reported through the sentinel surveillance system, integration of spatial and temporal entomological data, and application of predictive modelling within a large urban surveillance framework. These strengths support evidence-based targeting of vector control interventions and provide actionable insights for strengthening urban dengue surveillance.

### Conclusion and recommendations

In urban Surat, India, declining reported Dengue incidence (2016–2020) occurred despite persistently elevated larval indices, particularly in the South and South-East zones, and recurrent post-monsoon peaks, indicating sustained transmission risk. Adolescents and young adults bore the highest burden, while public-sector–dominant reporting indicates substantial underreporting from private providers. Weak associations between larval indices and Dengue incidence and moderate model discrimination (AUC = 0.687) confirm the limitations of single-indicator surveillance in dense urban settings.

Municipal programmes should implement zone-specific vector control during pre- and post-monsoon periods, mandate private-sector Dengue reporting, and target adolescents and young adults with focused risk communication and preventive interventions.

### Supporting information

**S1 Text. Table A. Annual Dengue case notification rates and entomological indices in Surat, India, 2016–2020.**
**Table B.** Zonal distribution of laboratory-confirmed reported Dengue incidences, notification rates, population characteristics, entomological indices, and reporting sectors in Surat, India, 2016–2020.
(DOCX)

### Acknowledgments

The authors sincerely thank the Surat Municipal Corporation for providing access to surveillance data and for their continued support of this study. We are especially grateful to the Multipurpose Health Workers and field surveillance teams for their dedicated efforts in data collection and vector surveillance, and for the valuable guidance and insights gained through collaboration with municipal public health staff.

### Author contributions

**Conceptualization:** Jigna D Gohil, Anjali M. Modi, Hiteshree C. Patel.

**Data curation:** Jigna D Gohil, Anjali M. Modi.

**Formal analysis:** Jigna D Gohil.

**Methodology:** Jigna D Gohil, Hiteshree C. Patel.

**Resources:** Anjali M. Modi.

**Supervision:** Anjali M. Modi.

**Validation:** Jigna D Gohil.

**Visualization:** Jigna D Gohil.

**Writing – original draft:** Jigna D Gohil.

**Writing – review & editing:** Jigna D Gohil, Anjali M. Modi, Jayendra K. Kosambiya.

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
