## [Decision Letter · Decision Letter 0]

15 Jan 2026

PGPH-D-25-01994

Spatiotemporal Patterns and Entomological Predictors of Dengue Transmission in Urban Surat, India (2016–2020): A Surveillance-Based Risk Modelling Study

Dear Dr. Jigna D Gohil,

Thank you for submitting your manuscript to PLOS Global Public Health. After careful consideration, we feel that it has merit but does not fully meet PLOS Global Public Health’s publication criteria as it currently stands. Therefore, we invite you to submit a revised version of the manuscript that addresses the points raised during the review process.

We look forward to receiving your revised manuscript.

Kind regards,

Muhammad Asaduzzaman, MD MPH MPhil

Academic Editor

Journal Requirements:

1. Please ensure that your Ethics Statement is available in its entirety at the beginning of your Methods section, under a subheading 'Ethics Statement'.

2. Please upload separate figure files in .tif or .eps format. Also, remove the figures from your manuscript file but keep the legends.

5. Some material included in your submission may be copyrighted. According to PLOS’s copyright policy, authors who use figures or other material (e.g., graphics, clipart, maps) from another author or copyright holder must demonstrate or obtain permission to publish this material under the Creative Commons Attribution 4.0 International (CC BY 4.0) License used by PLOS journals. Please closely review the details of PLOS’s copyright requirements here: PLOS Licenses and Copyright. If you need to request permissions from a copyright holder, you may use PLOS's Copyright Content Permission form.

Potential Copyright Issues:

a. Figure 1: Please confirm (a) that you are the photographer; or (b) provide written permission from the photographer to publish the photo(s) under our CC-BY 4.0 license.

b. Figure 3: please (a) provide a direct link to the base layer of the map (i.e., the country or region border shape) and ensure this is also included in the figure legend; and (b) provide a link to the terms of use / license information for the base layer image or shapefile. We cannot publish proprietary or copyrighted maps (e.g. Google Maps, Mapquest) and the terms of use for your map base layer must be compatible with our CC-BY 4.0 license.

Additional Editor Comments (if provided):

Reviewers' comments:

Reviewer's Responses to Questions

**Comments to the Author**

1. Does this manuscript meet PLOS Global Public Health’s publication criteria?

Reviewer #1: Yes

Reviewer #2: Yes

Reviewer #3: Partly

Reviewer #4: Yes

2. Has the statistical analysis been performed appropriately and rigorously?

Reviewer #1: Yes

Reviewer #2: Yes

Reviewer #3: Yes

Reviewer #4: Yes

3. Have the authors made all data underlying the findings in their manuscript fully available (please refer to the Data Availability Statement at the start of the manuscript PDF file)?

Reviewer #1: Yes

Reviewer #2: Yes

Reviewer #3: No

Reviewer #4: Yes

4. Is the manuscript presented in an intelligible fashion and written in standard English?

Reviewer #1: Yes

Reviewer #2: Yes

Reviewer #3: No

Reviewer #4: Yes

Reviewer #1: Reviewer Comments

Thank you for the opportunity to review this manuscript. The authors present a well-conducted surveillance-based study integrating laboratory-confirmed dengue case data with entomological and spatial information over a five-year period in an important urban setting. The topic is timely and relevant, and the manuscript aligns well with the scope of PLOS Global Public Health.

Overall, the study is technically sound, the statistical analyses are appropriate, and the conclusions are generally supported by the data presented. Key strengths include the use of laboratory-confirmed dengue cases, the integration of routine entomological surveillance, and the city-wide spatial analysis. The findings regarding persistent vector indices despite declining reported dengue incidence, spatial heterogeneity across zones, and underreporting from private health facilities are particularly valuable from a public health perspective.

That said, several areas would benefit from clarification and refinement to strengthen the manuscript and its interpretation:

Public health purpose of the regression model

The logistic regression model predicts moderate-to-high vector density (House Index ≥1%), which is programmatically relevant. However, the manuscript would benefit from a clearer explanation of why this outcome was selected and how the model is intended to be used operationally (e.g., early warning, prioritization of vector control activities, or surveillance planning). Clarifying this applied purpose would strengthen the public health relevance of the modelling component.

Interpretation of declining dengue incidence

The observed decline in dengue incidence over time—particularly during 2019–2020—is likely influenced by underreporting associated with COVID-19–related disruptions. I recommend consistently referring to these trends as “reported dengue incidence” and more explicitly emphasizing the role of surveillance artefacts when interpreting temporal patterns.

Model performance and interpretation

The model demonstrates moderate discrimination with high specificity but relatively low sensitivity and limited explanatory power. While this is acknowledged, the implications could be discussed more explicitly. Framing the model as a screening or prioritization tool—better suited for identifying lower-risk areas rather than precisely predicting hotspots—may help avoid overinterpretation.

Entomological indices and transmission dynamics

The discussion appropriately notes the weak and inconsistent relationship between larval indices and dengue incidence. This section could be strengthened by further acknowledging the limitations of larval indices as proxies for transmission risk, particularly in the absence of adult mosquito density measures, climatic variables, or human mobility data.

Private-sector reporting

The finding of substantial underreporting from private healthcare facilities is important. Additional discussion on possible contributing factors (e.g., reporting compliance, diagnostic access, or system incentives) and the implications for strengthening integrated urban surveillance would add value.

Minor language and stylistic edits—particularly in the Discussion—would improve clarity and flow but do not detract from the overall quality of the manuscript.

In summary, this is a solid and relevant study with clear potential for publication. With revisions addressing the points above, the manuscript would make a valuable contribution to the literature on urban dengue surveillance and control.

Reviewer #2: Section-by-Section Analysis

Title

Nothing special

Abstract

Adapt the abstract accordingly

Introduction

The background section is up to date and appropriate for the topic. The research gap is identified. However, the problem statement, research question, and hypothesis should have been clearly mentioned.

Methods

The study methods (including applicability/modelling) are reported in sufficient detail to allow for their replicability or reproducibility. However,

- the specific paradigm underpinning the study needs to be added: the post-positivist paradigm with quantitative design

- The study explores the temporal pattern. So, it is important to specify a ‘retrospective longitudinal design

- Details from lines 103 to 106 don’t have anything to do with the materials section. Move it to the introduction section (background)

The statistical analyses, control sampling, mechanism, and statistical reporting are appropriate and well described.

Results

The results presentation is not sufficiently effective to present the study findings.

- From lines 201 to 204, there is a logical issue. “ Diagnostic confirmation was achieved through detection of the non-structural protein (NS1) antigen in 66.0% of cases, dengue-specific immunoglobulin M (IgM) antibodies in 44.7%, and both markers in 10.8% of cases.”

- In Figure 2: What do you mean by Breatue Index in the legend? Isn’t the Breteau Index

- Temporal distribution must successively have at least two presentations of data: the first one is done (annual temporal distribution showing annual trends). The second must be the monthly distribution, showing the seasonal pattern (presenting cumulative data for each month). The description from lines 213 to 216 is not enough to demonstrate the seasonal pattern.

- Details from lines 217 to 220 are the authors’ comments, inappropriate in the results section. Please move your comments to the discussion section.

- From lines 224 to 227, please harmonize units (percentage or number). Don’t mix them (…in 64% of surveyed areas; however, 12 areas…)

- From lines 231 to 234, do not comment and interpret your results in this section

- In line 259, do not confuse gender with sex

Discussion

The interpretation of results and study conclusions is supported by the data and the study design. However,

- From lines 310 to 312, the rationale is not clear. The high rate reported from the public sector must be interpreted appropriately. Does active surveillance apply both in the public and private sectors? What is the relative size of public versus private health facilities in these areas? The analysis should have taken these elements into account.

- From lines 320 to 323, the argumentation is not logical. The authors showed a weak correlation between vector indices and case counts, while concluding that “Integrating epidemiological and entomological data is therefore essential for early warning and defining actionable thresholds for response

- Delimitations, limitations, and the strengths of the study are poorly described. Please detail factors that undermine the generalizability of the findings (delimitations), the constraints in data interpretation (limitations). Discuss all biases related to your study (selection, measurements, and interpretation biases) to improve transparency.

- References are not recent and diverse enough

Recommendations

Accepted with major improvement

Reviewer #3: Title: Spatiotemporal Patterns and Entomological Predictors of Dengue Transmission in Urban

Surat, India (2016–2020): A Surveillance-Based Risk Modelling Study

Comments to the authors

First of all, I would like to thank the editor for your invitation to review this paper. I appreciate the authors for their effort to conduct this manuscript too. The topic is very good and one of the public health importance issue in many countries.

Abstract part:

The authors would do better to include the objective of the study at the end of the background part of the abstract section.

What does it mean by "retrospective analysis"… in line 11? Is it to indicate the study design, data collection, or what? It is better to be clear. The design is not clear.

Methods part:

The method part is the blueprint of the main document so that it could be more likely good if it included the study design, the study population, the study period, the area, the way of data collection, the tool, and the software that was used for data entry and analysis.

It is not clear how the authors extract the data. How did the authors declare the associated factors and how did they check the assumptions of multicollinearity?

Results and conclusions:

How were the candidate variables selected for multivariable logistic regression? How did the authors check the presence of persistent vectors? The conclusion and recommendation are not in line with the results.

Main document

Introduction section:

Lines 86 and 87: The authors describe the idea without citation. The authors are better at showing the magnitude, severity, and consequences of the problem.

Methods part:

The design and study period are not clear, and the figure should be constructed in the standard format.

What is the importance of doing chi-square if the authors did logistic regression? If they did multivariable logistic regression, why did they not do bivariate logistic regression rather than doing chi-square?

Results section:

All of the figure captions are written above the figure. The figure captions should be written below the figure.

How was the age analyzed? Was it category data or continuous data? This description on table one and line 278 is not clear.

Discussion section:

It is shallow and not discussed well. It is not based on the right protocol of discussion in scientific research.

Conclusion and recommendations:

The authors did not include these sections.

Reviewer #4: Manuscript Review Report

I would like to commend you for this comprehensive and well-executed study on dengue transmission in Surat, India. The integration of spatiotemporal analysis, entomological surveillance, and predictive modelling provides invaluable insights into the dynamics of dengue transmission and the spatial hotspots in the city. The findings from this study offer significant potential to enhance dengue control measures, particularly in rapidly urbanising regions. Below are analytical comments intended to refine and strengthen your manuscript. These suggestions focus on clarifying methodological details, deepening the interpretation of findings, and enhancing policy relevance.

1. Your abstract outlines the study's objectives, methods, and findings; the implications for dengue control could be made more actionable, particularly in terms of targeted interventions.

2. Your introduction discusses the spatiotemporal nature of dengue transmission, the broader urbanisation and climate change trends that exacerbate dengue risk in tropical cities could be framed more strongly within global health policy contexts.

3. While vector indices are valuable, their predictive power can be limited, as evidenced by the lack of correlation between vector density and disease incidence in some years. Providing more context on the limitations of these indices would strengthen the interpretation of the findings. Your methods section could benefit from a more explicit explanation of the limitations of using vector indices (House Index, Container Index, Breteau Index) for predicting dengue incidence.

4. Clarify the reasoning behind choosing logistic regression as the primary modelling technique, particularly in relation to other potential approaches (e.g., spatiotemporal models or Bayesian approaches). While logistic regression is appropriate for identifying predictors of high vector density, the study may benefit from more complex models that account for spatial autocorrelation and temporal dependencies, which may better capture the dynamics of disease spread.

5. You mentioned that 85% of cases were reported through the public sector; the extent of underreporting by private providers could be explored in more detail, particularly in relation to surveillance gaps in urban areas.

6. You acknowledged the moderate discriminatory ability of the logistic regression model (AUC = 0.688), but a more detailed **discussion of why the model has low sensitivity (35.9%) despite high specificity would provide further clarity on the model’s utility for public health purposes.

7. Your conclusion about sustained transmission risk despite declining reported cases is well-argued. However, further emphasis on integrating socio-economic factors (e.g., migration patterns, urban poverty) into predictive models could significantly improve disease control.

8. Although the study provides valuable insights into spatial transmission patterns and entomological predictors, the conclusion could be more actionable by outlining specific dengue control strategies for urban health systems.

**Do you want your identity to be public for this peer review?** For information about this choice, including consent withdrawal, please see our Privacy Policy

Reviewer #1: No

Reviewer #2: No

Reviewer #3: No

Reviewer #4: No

---

## [Decision Letter · Decision Letter 1]

17 Feb 2026

Spatiotemporal Patterns and Entomological Predictors of Dengue Transmission in Urban Surat, India (2016–2020): A Surveillance-Based Risk Modelling Study

PGPH-D-25-01994R1

Dear Jigna D Gohil,

We are pleased to inform you that your manuscript 'Spatiotemporal Patterns and Entomological Predictors of Dengue Transmission in Urban Surat, India (2016–2020): A Surveillance-Based Risk Modelling Study' has been provisionally accepted for publication in PLOS Global Public Health.

Best regards,

Muhammad Asaduzzaman, MD MPH MPhil

Academic Editor

Reviewer Comments (if any, and for reference):

Reviewer's Responses to Questions

**Comments to the Author**

Reviewer #2: All comments have been addressed

Reviewer #4: All comments have been addressed

publication criteria?

Reviewer #2: Yes

Reviewer #4: Yes

3. Has the statistical analysis been performed appropriately and rigorously?

Reviewer #2: Yes

Reviewer #4: Yes

4. Have the authors made all data underlying the findings in their manuscript fully available (please refer to the Data Availability Statement at the start of the manuscript PDF file)?

Reviewer #2: Yes

Reviewer #4: Yes

5. Is the manuscript presented in an intelligible fashion and written in standard English?

Reviewer #2: Yes

Reviewer #4: (No Response)

Reviewer #2: Congratulations!

Reviewer #4: The authors have effectively addressed most of the concerns raised in the initial review. The manuscript is now more comprehensive and transparent, with clearer explanations of methodology, stronger integration of findings, and actionable recommendations for public health interventions. The changes made significantly improve the manuscript's clarity, relevance, and policy implications. I recommend that the manuscript be accepted for publication, as the authors have addressed the feedback in a thoughtful and thorough manner.

**Do you want your identity to be public for this peer review?** For information about this choice, including consent withdrawal, please see our Privacy Policy

Reviewer #2: **Yes:** Dibeka Baman

Reviewer #4: No
